# Preference Explanation and Decision Support for Multi-Objective Real-World Test Laboratory Scheduling

## Abstract

Complex real-world scheduling problems often include multiple conflicting objectives. Decision makers (DMs) can express their preferences over those objectives in different ways, including as sets of weights which are used in a linear combination of objective values. However, finding good sets of weights that result in solutions with desirable qualities is challenging and currently involves a lot of trial and error. We propose a general method to explain objectives' values under a given set of weights using Shapley regression values. We demonstrate this approach on the Test Laboratory Scheduling Problem (TLSP), for which we propose a multi-objective solution algorithm and show that suggestions for weight adjustments based on the introduced explanations are successful in guiding decision makers towards solutions that match their expectations. This method is included in the TLSP MO-Explorer, a new decision support system that enables the exploration and analysis of high-dimensional Pareto fronts.

## 1 Introduction

In many complex real-world optimization problems, trade-offs are required as optimizing for one desired preference comes at a cost to another. The area of multi-objective optimization deals with such problems, where several objectives should be optimized at once. A popular approach to determine the best solution to such problems is to capture the preferences of domain experts and decision makers (DMs) between those objectives numerically in such a way that solutions can be totally ordered and optimizing with respect to this order. Several different approaches exist, such as weighted linear combinations of objective values (Pajer et al. 2017), scalarizing functions using reference points (Misitano et al. 2022), and more. However, expressing and defining these preferences is a difficult task in itself. The consequences of any particular preference structure (e.g. a set of objective weights) on the solutions are often hard to predict, especially where the range of possible solutions is initially unknown. Further, DMs may want to understand how the solutions that are found by automated solvers have been generated and require solvers to explain the decisions they have made. This information can be provided in the form of (approximated) Pareto fronts (see Section 2): Populations of solutions such that an improvement in one objective is possible only via an increase in another. Unfortunately, Pareto fronts for problems with more than two objectives are hard to visualize and understand, which makes it necessary to develop decision support tools that assist the DM in the analysis and exploration of alternative solutions.

This paper deals with a complex real-world scheduling problem, the Test Laboratory Scheduling Problem (TLSP). The TLSP was first described in (Mischek and Musliu 2018, 2021) in order to model the requirements of scheduling activities in industrial test laboratories, where a large number of tests have to be performed subject to complex demands regarding time windows, ordering of tests, as well as qualification and certification of several different resources. The original formulation of the TLSP used a weighted linear combination of several objectives in order to determine the quality of otherwise feasible schedules. Previous works proposing solution approaches for the TLSP (e.g. (Mischek, Musliu, and Schaerf 2023; Danzinger et al. 2023; Geibinger, Mischek, and Musliu 2021)) assumed uniform weights of 1 in their evaluations, with a discussion on the difficulty of finding "good" weights in practice (i.e. ones that lead to solutions that are desirable for the DMs) given in (Danzinger et al. 2023). Weights are roughly estimated based on practical experiences with previous scheduling runs, which can lead to counter-intuitive or undesirable situations, where the solver produces solutions that are optimal with respect to the given weights, but do not match the expectations and preferences of the DM. In these situations, it is important for the DM to be able to explore alternative solutions and analyze where the weights they have set previously lead to these undesired results.

The contributions of this paper are as follows:

- We describe a multi-objective solution algorithm for the TLSP, using Pareto-Simulated Annealing (PSA) combined with Very Large Neighborhood Search (VLNS).

- We propose a new method to calculate the contribution of objectives on each others' value that is based on Shapley regression values. From these Shapley values we compute suggestions for objective weight adjustments and show that following the suggestions is beneficial for improving the value of desired target objectives.

- We introduce TLSP MO-Explorer, a graphical decision support system that allows DMs to interactively explore a set of populations of solutions for the TLSP, such as

those generated by the PSA algorithm. The system also includes the suggestions for weight adjustments computed via Shapley regression values, which provide insights into the interaction between different objectives. A short user study with domain experts from our industrial partner, who provided qualitative feedback on the system prototype and a quantitative assessment of its usability, shows the usefulness of this system in assisting with real-world scheduling tasks.

This paper is structured as follows: The next section introduces theoretical background information and definitions, including a description of the TLSP. We introduce our multi-objective solution approach for the TLSP in Section 3. Section 4 contains the description of our proposed explanation approach using Shapley regression values. We demonstrate on the example of the TLSP that following the suggestions for weight adjustments derived from these values is a promising strategy to reduce the value of certain target objectives. In Section 5 we describe the TLSP MO-Explorer, as well as the user study with domain experts. Finally, the last section contains our conclusions and lessons learned, as well as an outlook on future research directions.

## 2 Background

### The Test Laboratory Scheduling Problem

We provide here a summary of the Test Laboratory Scheduling Problem (TLSP). The full formal problem description is included in the appendix (Section A) and can also be found in (Mischek and Musliu 2021).

The TLSP is a project scheduling problem, where the solver has to provide a schedule for multiple *projects*, each containing several *tasks*. Each task has a duration, a time window in which it must be performed, requires multiple different and heterogeneous resources, and can be performed in one of several possible modes. Additional constraints prescribe a precedence order between tasks, limit which units of a resource are suitable for a task and link tasks that should be assigned the same resource units.

However, tasks are not scheduled directly, but must first be grouped into larger units called *jobs*, which are then assigned a time interval, execution mode, and resource units. Each job derives its properties from the tasks it contains, under the assumption that tasks within a job are performed sequentially, but without any defined order. As a consequence, a job must fulfill all requirements of its tasks for its whole duration, which is defined as the sum of the durations of its tasks, plus an additional setup time. For example, a job can start only after the latest release date among all its tasks, and after the end of all other jobs that contain prerequisite tasks of at least one of its tasks. Similarly, it must be assigned enough units of a resource to cover the highest demand among its tasks and resource units that are suitable for all its tasks can be assigned to it.

A combination of different objectives determines the quality of a schedule. The default version of the TLSP considers the following objectives:

1. The number of jobs should be minimized.

2. The employees assigned to a job should be taken from a further subset of the suitable employees, the *preferred* employees.

3. The number of different employees assigned to each project should be minimized.

4. The internal due date for each job should be observed, which is typically before the deadline.

5. The total completion time (from the start of the first job to the end of the last) of each project should be minimized.

Additional objectives have been described for practical applications (Danzinger et al. 2023), however in this work we restrict our discussion to those objectives already present in the original formulation.

### Dominance and Pareto fronts

An important concept of multi-objective optimization is that of *dominance* between solutions: Given evaluation functions $f_i$ for each objective $i$, $\mathbf{x}$ *dominates* $\mathbf{y}$ if $f_i(\mathbf{x}) \leq f_i(\mathbf{y})$ for all objectives and $f_i(\mathbf{x}) < f_i(\mathbf{y})$ for at least one objective. The set of *Pareto optimal* solutions are those that are not dominated by any other solution. Their corresponding set of objective vectors is called the *Pareto front*. In practice, computing exact Pareto fronts is typically computationally infeasible, so approximations are used instead.

### Shapley values

Shapley values (Shapley 1952) were originally devised as a measure for the contribution of individual players to the total payout of a coalitional game: Given a set $N$ of $n$ players and an evaluation function $v : 2^N \to \mathbb{R}$ assigning each subset of players a total payout, the Shapley value of player $i$ is defined as

$$\varphi_i = \sum_{S \subseteq N \setminus \{i\}} \frac{|S|! \, (n - |S| - 1)!}{n!} (v(S \cup \{i\}) - v(S)) \quad (1)$$

They can be shown to guarantee important theoretical properties, such as *local accuracy*, *missingness*, and *consistency* (Lundberg and Lee 2017), and have been used as the basis of several popular formalisms to explain the influence of input parameters of machine learning models on the model output. For example, Shapley regression values (Lipovetsky and Conklin 2001) are computed by directly applying equation 1 with the input features as players and $v(C)$ being the output of a model trained on feature subset $C$. One major drawback of this approach is that it requires the training of a separate model for each subset of the input features, as most machine learning techniques are unable to work with only a subset of the original training features.

SHAP values (Lundberg and Lee 2017) aim to circumvent the above problem by replacing the computation of partial models (i.e. models over a subset of features) with a conditional expectation function on the original model, preserving the same theoretical properties. Since the exact computation of SHAP values is still computationally expensive, in particular for models with a large number of features, both model-agnostic and model-specific approximation methods exist.

### Explainable interactive multi-objective optimization

The setting of explainable interactive multiobjective optimization (XIMO) (Misitano et al. 2022) is as follows: We are given a multi-objective optimization problem with $k$ objectives and a certain preference structure $\mathbf{z} \in \mathbb{R}^k$ over those objectives that allows us to compute a single "best" solution relative to this structure. Originally, this setting was defined only for preference structures taking the form of *reference points* in objective space. However, there are also other widely-used ways of expressing preferences between objectives (Miettinen 1999), including vectors of *objective weights* used in a linear combination of penalties.

A DM provides an initial estimate $\mathbf{z}^0$ of those preferences. This estimate is passed to a black-box optimizer $\mathcal{BB}$ which produces a solution with an objective vector $\mathbf{x}^0 = \mathcal{BB}(\mathbf{z}^0)$ based on the given preferences[1]. While $\mathbf{x}^0$ is the best solution (found) relative to $\mathbf{z}^0$, it may still not match the expectations of the DM.

The goal in XIMO is to interactively assist the DM in achieving a desirable solution by explaining *how* and *why* the returned solution is the best for the provided preferences and give suggestions on how those preferences can be adjusted in order to achieve the DM's goals (e.g. to reduce the value of certain objectives). The DM can then provide an updated preference structure $\mathbf{z}^1$, which yields a new solution $\mathbf{x}^1 = \mathcal{BB}(\mathbf{z}^1)$, and repeat until they are satisfied.

The R-XIMO method (Misitano et al. 2022) translates the formalism of using SHAP values for explaining machine learning models to this setting, where preferences between objectives are given as reference points. It is then possible to compute SHAP values $\phi_{ij}$ which indicate the contribution of component $j$ of the given reference point towards the value of objective $i$ in the resulting solution. The authors of R-XIMO use these SHAP values (estimated using Kernel SHAP) to provide suggestions on how to interactively modify the reference point in order to improve the value of one particular objective. One limitation of that approach is that it requires an additional set of background values, representing typical or aggregate vectors whose values are used in place of masked objectives. In R-XIMO, the approximated Pareto front itself is used as background data, which is empirically shown to work well with reference points, but is unsuitable for other types of preference structures.

## 3 Multi-objective solving techniques for the TLSP

Previous solution approaches to the TLSP have focused on searching for near-optimal solutions with respect to a single set of objective weights. To provide decision makers with a set of diverse solutions, we instead need methods that approximate the Pareto-front along its full length.

For this, we propose a Pareto Simulated Annealing (PSA) for TLSP based on the general PSA framework (Czyżżak

---

[1]For simplicity and computational efficiency, we can assume that the optimizer has access to a (precomputed) set of non-dominated solutions close to the Pareto front and chooses its results from this set.

and Jaszkiewicz 1998). PSA keeps a population $S$ of solutions of size $p$ and performs a local search procedure similar to Simulated Annealing (SA) to each of them individually. At each iteration, this procedure generates a random new solution $\mathbf{y}$ from the neighborhood of its current solution $\mathbf{x}$. The acceptance of this neighboring solution is determined by a multi-objective version of the Metropolis criterion in SA $P(\mathbf{x}, \mathbf{y}, T, \mathbf{\Lambda})$, which also depends on the current temperature $T$ and a vector of weights $\mathbf{\Lambda} = (\lambda_1, \ldots, \lambda_k)$ (for $k$ objectives). In this work, we used the SL rule, which is defined as follows:

$$P(\mathbf{x}, \mathbf{y}, T, \Lambda) =$$
$$\min(1, \exp(\frac{\sum_{j=1}^{J} \lambda_j(f_j(\mathbf{x}) - f_j(\mathbf{y}))}{T})) \quad (2)$$

Since intermediate solutions can be infeasible, we add the number of hard constraint violations as an additional 'objective' with a fixed weight of 100.

Each solution $\mathbf{x}$ in the population has its own weight vector $\mathbf{\Lambda}^{\mathbf{x}}$ which determines the direction in the objective space in which the solution tends to move. These weight vectors are initialized randomly and after each iteration, the weight vector of each solution $\mathbf{x}$ is updated[2] based on its closest other solution $\mathbf{x}' \in S \setminus \{\mathbf{x}\}$:

$$\lambda_j^{\mathbf{x}} = \begin{cases} \alpha \lambda_j^{\mathbf{x}} & \text{if } f_j(\mathbf{x}) \leq f_j(\mathbf{x}') \\ \lambda_j^{\mathbf{x}}/\alpha & \text{if } f_j(\mathbf{x}) > f_j(\mathbf{x}') \end{cases} \quad (3)$$

This increases the probability of each solution to move away from its closest neighbor, which tends to disperse the population across the whole region of (near-)Pareto-optimal solutions. Weight vectors are normalized to a total weight of 1 after each update. In addition, we enforce a minimum weight of 0.001 for each objective, to avoid solutions where some objectives are completely ignored.

The neighborhood definitions and selection probabilities, as well as the cooling scheme were adapted from the (single-objective) SA approach described in earlier work (Mischek, Musliu, and Schaerf 2023).

To generate the initial population, we first used randomly generated solutions. However, this approach turned out to have difficulties finding feasible solutions, particularly for large instances. Therefore, we propose a hybridization of PSA with the VLNS approach from (Danzinger et al. 2023): PSA with VLNS initialization (PSA-VI) initially runs VLNS on each solution in the population (VLNS itself starts out from a greedily constructed initial solution). Once it finds a feasible solution, it switches to PSA for that solution for the rest of the run.

To evaluate our approach, we compared PSA-VI with pure PSA, a variant that keeps running VLNS even on feasible solutions (PVLNS) while also updating the objective weights after each move as in (3), and the two single-objective algorithms SA and VLNS (using a linear combination of objective values with uniform weights and keeping track of all non-dominated solutions found during the

---

[2]The signs are reversed compared to (Czyzżak and Jaszkiewicz 1998), as we deal with a minimization problem

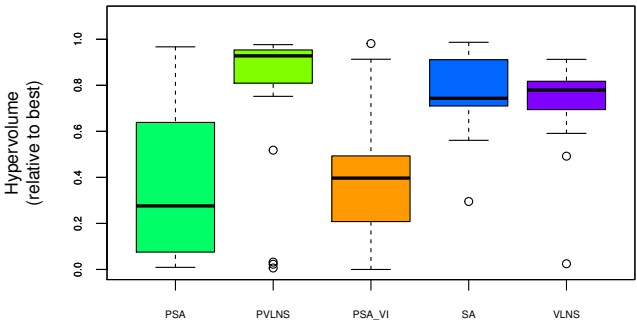

Figure 1: Hypervolumes spanned by the solution sets produced by different approaches, relative to the biggest hypervolume found in any run. Hypervolumes for each instance were calculated relative to the same point for all approaches.

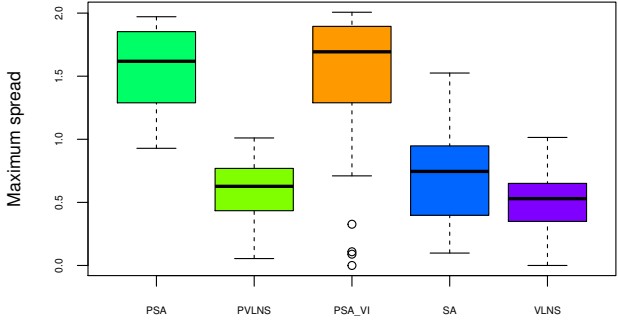

Figure 2: $M_3^*$ maximum spread measure achieved by the different methods. Objective values were normalized to the interval [0,1] for each instance and objective.

search). We evaluated these approaches on a set of 33 publicly available standard benchmark instances[3]. These instances include 3 real-world instances as well as 30 randomly generated ones. They contain between 13 and 1573 tasks to be scheduled and 150.9 resource units on average, split across up to 8 different groups (see (Mischek and Musliu 2021) for more details on the benchmark instance set). Each algorithm was run 5 times on each instance with a timeout of 2 hours per run, using up to 6 cores in parallel, with $p = 20$ for PSA(-VI) and PVLNS.

The results show that PSA-VI achieves a good balance between solution quality and diversity compared to the other approaches: It found feasible results on $84\%$ of all runs, significantly more than both PSA ($73\%$) and VLNS ($75\%$), while roughly on par with PVLNS ($82\%$) and SA ($85\%$). Of note is that while SA and VLNS are single-threaded, they had the significant advantage of being able to spend all their time on a single solution, while PSA-(VI) and PVLNS had to spread their efforts over all 20 solutions.

While the hypervolumes (Fonseca, Paquete, and Lopez-Ibanez 2006) spanned by the solution sets found by PVLNS, SA, and VLNS are larger (Figure 1), the diversity of those solutions (Figure 2), measured by the maximum spread measure ($M_3^*$, (Zitzler, Deb, and Thiele 2000)) is clearly bigger for PSA(-VI). Detailed experimental results are included in the appendix (Section B).

## 4 Explanations for multi-objective scheduling problems using Shapley values

In this paper, we generalize the XIMO setting to multi-objective problems with $k$ objectives and any preference structure of the form $\mathbf{z} \in \mathbb{R}^k$. Let $N = \{1, \ldots, k\}$ be the set of all objectives. We only require that the black-box optimizer $\mathcal{BB}$ using that structure is also able to work with subsets of those objectives. In other words, a function $\mathcal{BB}_S$ must exist for all $S \subseteq N$ that selects the best solution $\mathbf{x}_S$ according only to the objectives in $S$ (and the corresponding components of $\mathbf{z}$). Natural definitions for such partial optimizers are easy to find for many popular ways to express preference

---
[3]https://dbai.tuwien.ac.at/staff/fmischek/TLSP/

structures: For example, for problems using objective weight vectors such as the TLSP (i.e. $\mathcal{BB}$ selects the solution $\mathbf{x}$ that minimizes $\sum_{i \in N} z_i f_i(\mathbf{x})$), we can simply treat the weight of all absent objectives as 0. For problems using reference points (like those treated by R-XIMO), we can similarly apply the scalarization function only over those objectives that are present and ignore all others.

With such a structure, we are able to use Shapley regression values directly instead of SHAP values, as we can now define the evaluation function $v$ in Equation 1 as follows for a target objective $t \in N$ and preference vector $\mathbf{z}$:

$$v_t(S) = f_t(\mathcal{BB}_S(\mathbf{z})) \qquad (4)$$

If we evaluate (1) for each $t$ and each $i \in N$, we get a matrix of Shapley values, where entry $\varphi_{ti}$ indicates the contribution of the $i$th component of $\mathbf{z}$ to the value of objective $t$ in the solution $\mathbf{x}$ returned by the optimizer. These values can then be used to provide insights to the DM on the relationship between objectives and provide suggestions on modifications to the provided preference structures to achieve the DM's goals, as we demonstrate on a case study below.

Since practical multi-objective optimization problems typically don't have hundreds of objectives, the Shapley values can also be calculated exactly in reasonable time, without having to rely on approximation methods.

### Application to the TLSP

We applied our approach to the TLSP using a similar experimental setup to the one used to evaluate R-XIMO (Misitano et al. 2022): We first generated approximations of the Pareto fronts for all 33 instances, using PSA-VI (see Section 3). After discarding the 5 instances for which no feasible solution was found, as well as one for which the returned solution set contains only a single solution, we have non-dominated solution sets for 27 instances, with at least 2 and up to 1037 solutions per instance, with an average of 348.97.

For each experiment, we start out with an initial vector of weights $\mathbf{w}^0$. Designating each of the five objectives, one after the other, as the *target* $t$, we compute Shapley regression values $\varphi_{ti}$ of all objectives $i$ with respect to those weights and the target. From that, we designate the objective other than the target with the highest value (i.e. the most inhibiting

| Strategy | Description |
|----------|-------------|
| A | Improve target, impair rival |
| B | Improve target only |
| C | Improve target, impair random other (not the target or rival) |
| D | Impair rival only |
| E | Impair random other (not the target or rival) |

Table 1: The five strategies used in the evaluation, as in (Misitano et al. 2022)

.

other objective) as the *rival*. As was done for R-XIMO, we compare five strategies on which objective to improve (i.e. increase its weight by a factor of $\gamma$), and which to impair (i.e. decrease its weight by a factor of $\gamma$). The five strategies are listed in Table 1. Applying strategy $S$ results in a new weight vector $\mathbf{w}^1_S$. We then compare the objective value of the target in the best solution relative to $\mathbf{w}^1_S$ with that of the best solution relative to $\mathbf{w}^0$. Since some strategies behave nondeterministically, we repeated each experiment 10 times, for a total of 1350 applications of each strategy per configuration (27 instances times 5 objectives times 10 runs).

To generate the initial weights $\mathbf{w}^0$, we compare two different approaches: In the *uniform* approach, we set a uniform weight of 1 for each objective. In the *exponential* approach, we assign a weight of $10^i$, with $i$ chosen uniformly randomly between 0 and 4 (inclusive) to each objective. The latter approach shows how the explanations deal with large differences between objective weights and is also similar to the weights currently in use in practice in the lab of our industrial partner. For the parameter $\gamma$, we compare the values 1.5, 2, and 10 (based on typical weight adjustments used by our industrial partner in practice).

The solution sets and the source code for our evaluation are included in the supplemental material.

**Experimental results**   Figure 3 shows for each combination of initial weight setting and update factor $\gamma$ the results of applying either of the five strategies from Table 1. For the purpose of this figure, a successful application of a strategy is one where the objective value of the target objective was reduced, while a failure indicates that the value of the target objective actually increased due to the change in weights. A neutral case is one where the target objective value remained the same, either because the best solution did not change or because the new best solution had the same value in the target objective. For successful and neutral adjustments, the darker shaded area additionally shows those results that are the best achievable out of all possible adjustments (consisting of up to one improved and up to one impaired objective).

The figures show that following the suggestions derived from the Shapley regression values (Strategy A) clearly provided the best results compared to the other strategies. It had both the highest rate of successful adjustments (39.4% of all experiments) and was the most likely to result in the best possible improvement (81.4%). Notably, both components of the strategy (improving the target and impairing the ri-

val) contribute to this success: Strategies improving the target (A, B, C) consistently yield better results than strategies that do not improve any objective (D, E) and strategies impairing the rival (A, D) yield better results than strategies that impair no or a random other objective (B, C, E). Unsurprisingly, smaller values for $\gamma$ result in lower chances to find an improvement.

Having an exponential distribution of initial weights also resulted in more neutral outcomes compared to the uniform case: If the target already has a much higher weight than the other objectives, then further increasing it is unlikely to have much effect. On the other hand, if another objective has a much higher weight than the rest, it will dominate the search for the best solution and again make changes to other objective weights less impactful unless they come close or surpass this high weight.

The average magnitude of change in the target objective value for each experiment is shown in Figure 4. Again, following Strategy A results in a higher reduction of target objective value than any of the other strategies. As expected, higher reductions are achievable with bigger weight update steps. Interestingly, the average reductions are smaller for uniform weight distributions than for randomly exponentially distributed ones, despite the smaller success rate. This indicates that where improvements are possible (e.g. when the highest weighted objective changes), they are far larger than in the uniform case.

## 5   Visualization and Interaction

While two- and to a lesser extent also three-dimensional Pareto fronts can directly be displayed visually, this is no longer the case for four or more dimensions. Much work has been done on visualization techniques for such high-dimensional data sets (Lotov and Miettinen 2008), which can be classified into three main approaches (Dimara, Bezerianos, and Dragicevic 2018): *Dimensionality reduction* strategies try to collapse the high-dimensional space into fewer dimensions, and include e.g. self-organizing maps (Obayashi, Jeong, and Chiba 2005), filtering (Yang et al. 2003), or principal component analysis (Huang et al. 2017). *Non-geometric visualization techniques* are often icon-based (Fuchs et al. 2016) or exploit structural relations between objectives (Keim and Kriegel 1996). Finally, *lossless geometric projection* strategies attempt to preserve the raw information of each data point. These include e.g. tabular methods (Gratzl et al. 2013) or scatterplot matrices (Emerson et al. 2013). For our use case, a lossless approach is important because DMs need to able to see the actual objective values of each solution.

One of the most popular lossless visualization approaches are parallel coordinate plots (PCP) (Bagajewicz and Cabrera 2003), where the different dimensions or objectives are represented by parallel axes. Each solution is then plotted as a single line in that graph, whose y-value at each axis corresponds to the solution's objective value of the objective associated with the axis. These plots can then be enhanced with interaction tools to support filtering, inspection, ranking, and selection of solutions, as was done e.g. in the WeightLifter system (Pajer et al. 2017).

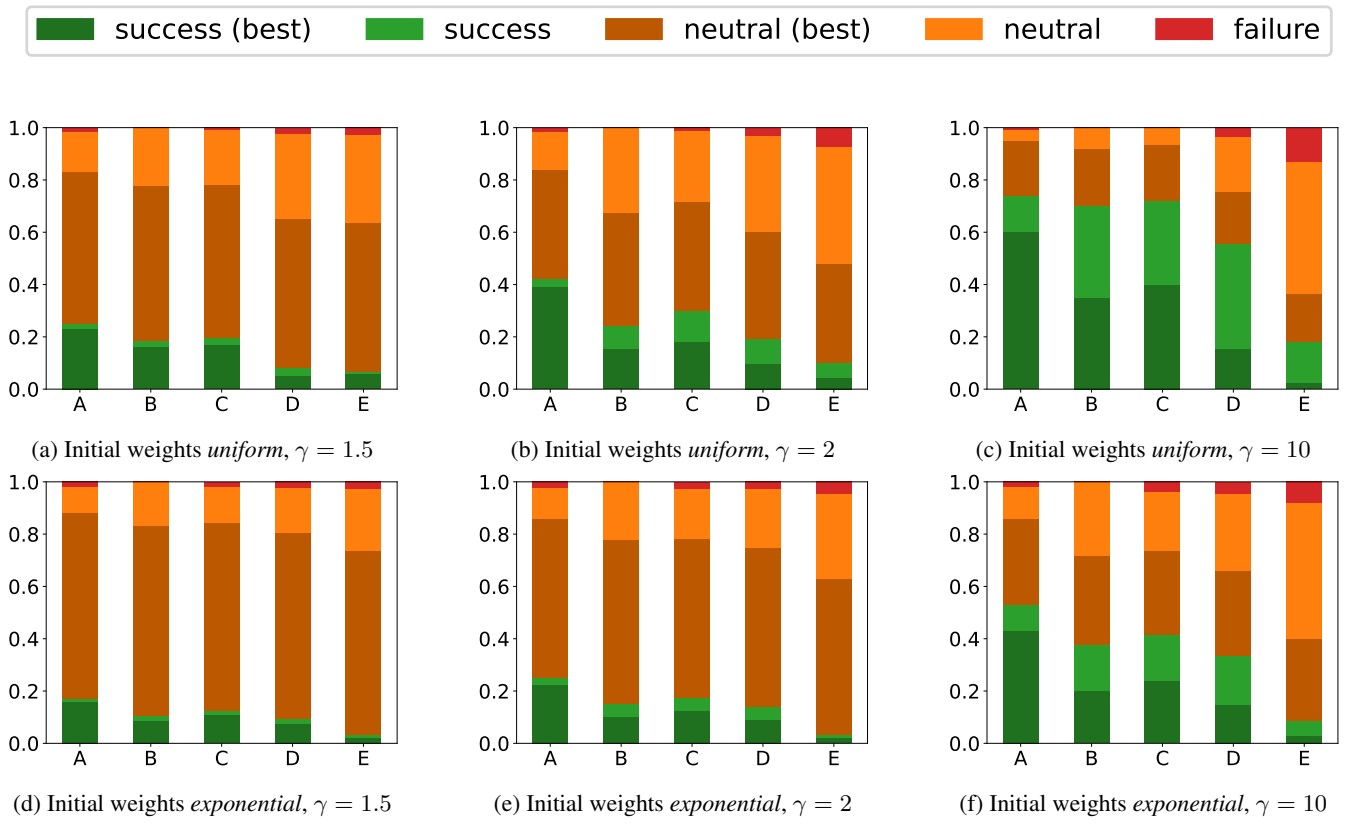

Figure 3: Success, neutral, and failure rates observed for each strategy. The darker shaded areas show those results where following the strategy resulted in the best result achievable by improving up to one objective and impairing up to one other.

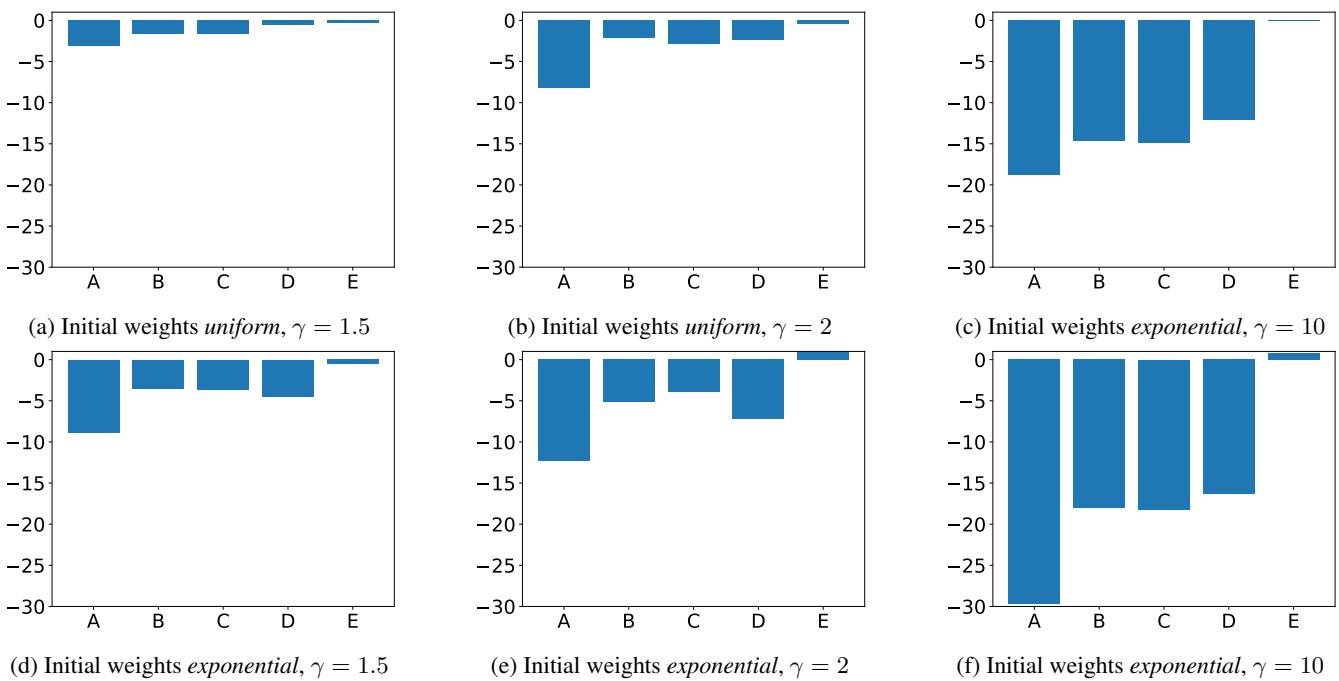

Figure 4: Average change in objective value achieved for each strategy.

Our visualization and exploration tool for the multi-objective TLSP proposed in this work (TLSP MO-Explorer) also uses a PCP, whose design and interaction tools are based on the PAVED system for the analysis of engineering design alternatives (Cibulski et al. 2020). We extend those tools with the explanations and suggestions for objective weight setting provided via Shapley value analysis. The prototype was implemented in Java 11, using the Swing toolkit to build the GUI and JFreeChart[4] for the PCP.

The main view of the TLSP MO-Explorer is shown in Figure 5. The top half shows the PCP, while the lower half contains interaction tools and additional information regarding individual objectives.

### Selection and filtering

Two important tasks are the selection of interesting solutions and the removal of undesirable solutions.

The best solution relative to the current objective weights (b) is always plotted on top in orange. In addition, users can select individual solutions (shown in blue) for further inspection or export by directly clicking on the chart. Solutions with undesirable values for some objectives can be greyed out via the value range controls for each objective (a), which grey out and move to the background all solutions that fall outside the specified range.

### Detailed inspection of solutions

While the PCP provides an overview over the objective values of each solution, DMs may wish to inspect certain solutions in more detail.

Such detailed views (c) can be expanded for each objective.These views provide tabular data for the solution, including penalty values for each individual project and additional information relevant to the chosen objective. For example, the details table for the 5 (Project completion time) displays the actual start and end dates of each project.

In addition, individual solutions can also be exported (f) in various formats to allow for offline inspection or upload of a schedule into the lab management system.

### Objective interaction and weight setting

Several tools assist the DM in studying the interaction between different objectives and help with determining appropriate objective weights for future solution processes.

They can select any objective to be *focused* (e): In the chart, solutions are color-coded according to their value in the focused objective, with a gradient from high values in red to lower values in green. This provides a visual indication of correlations between the objectives, such as the negative correlation between objective 5 (Project completion time) and objectives 1 (Number of jobs) as well as 3 (Same resource).

The last row (d) uses the Shapley regression values described in Section 4 to describe objective interaction effects. The arrows indicate the impact of other objectives on the focused objective at their current weight (downward arrow: decreasing effect, upward arrow: increasing effect). Those objectives with the highest increasing and lowest decreasing

---

[4]https://www.jfree.org/jfreechart/

effect are additionally emphasized with a double arrow. The actual Shapley values are available as a tooltip.

The current objective weights (b) can be edited to see how different preferences change the solution regarded as best, as well as the interaction effects between objectives. The effect indicators in the last row provide guidance on how the weights should be edited to reduce the value of the focused objective: The weight of objectives with a decreasing effect (downward arrows) should be raised, while the weight of objectives with an increasing effect (upward arrows) should be lowered, particularly the two most influential objectives.

### User interview

To evaluate the usefulness of our prototype in practice, we conducted a demonstration and interview session with domain experts from our industrial partner, DM1 and DM2. Both are familiar with the TLSP and routinely use solvers based on weighted linear combinations of the objectives for daily scheduling in the lab.

As preparation, we generated an approximated Pareto front for an instance of the TLSP taken from the lab's database. After a brief overview on the TLSP MO-Explorer prototype and the PCP, we asked the DMs to perform several tasks within the system.These tasks included filtering solutions based on their values in two specific objectives, selecting an interesting solution and inspecting additional details for it, adjusting the weights in order to decrease the value of one specific objective in the best solution relative to the current weights (with a focus on interactions between the chosen objective and others), and finally exporting solutions they found interesting for further processing. While they were exploring the tool, we asked the DMs to think aloud and took notes on their comments. Afterwards, we conducted a short interview regarding their experience and asked them to rate the prototype's usability on the Software Usability Scale (SUS) (Brooke 1996).

**Insights and results**   During exploration and the interview, both DMs reported that the visualization and analysis of alternative solutions (compared to the single "best" solution provided by existing solvers) was helpful for them. DM1 found that this capability would be particularly interesting for short-term scheduling (i.e. for the next few weeks), as finding good schedules is particularly challenging there and trade-offs are unavoidable. DM2 remarked that they would even prefer to employ the multi-objective exploration by default for all scheduling tasks, as the graphical overview of solution alternatives and the option to analyse the effect of shifting priorities between objectives proved very useful for them. They also remarked that the overall value ranges of individual objectives provide valuable insights for judging what kind of solutions can be achieved at all.

While solving the given tasks, both DMs asked for clarification regarding the visualization of the objective interaction effects with upwards and downwards arrows. After explaining the meaning behind the arrows, the DMs agreed that this feature would be helpful in adjusting the weights to achieve desirable solutions. Other features that required explanation were the selection of individual objectives to focus on, and

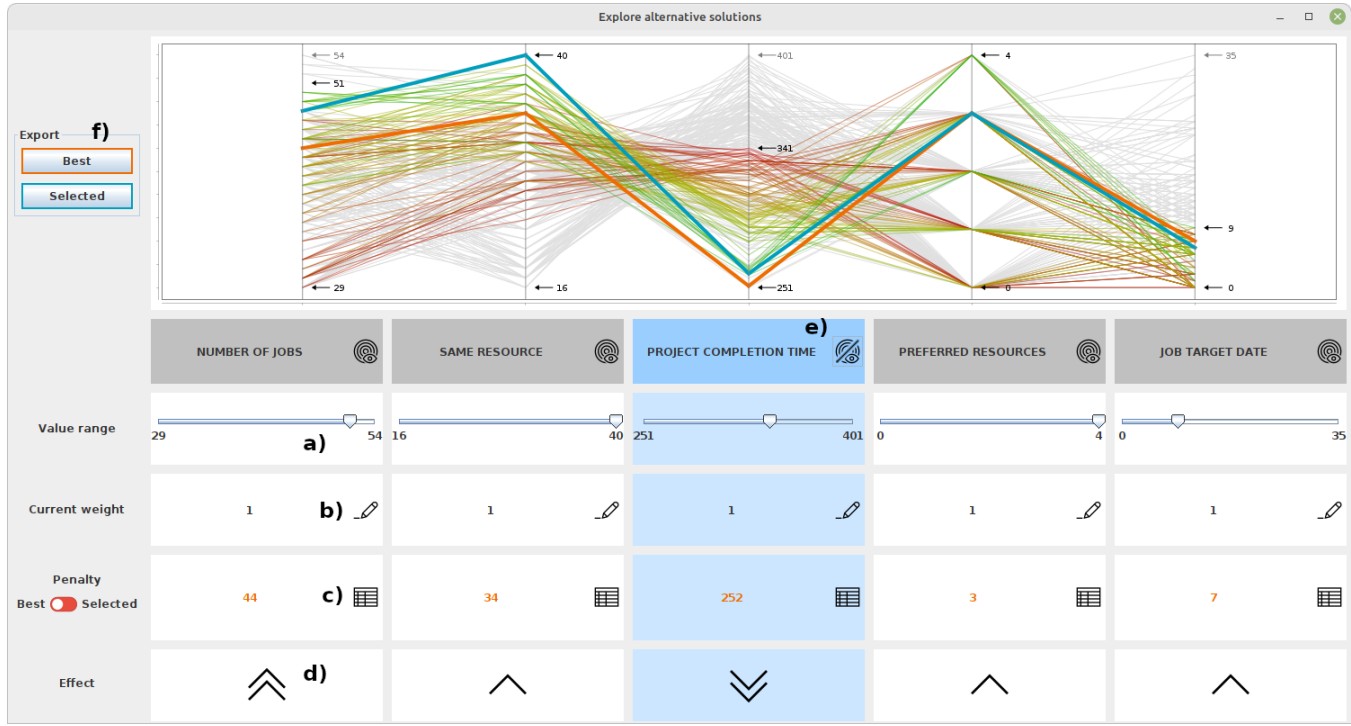

Figure 5: Main view of the TLSP MO-Explorer. Each vertical axis corresponds to the objective listed below it, while solutions are represented as lines in the plot. The columns below each objective provide interaction tools and additional information.

the effect of the value range filters. DM1 also remarked that without the color gradient from having a focused objective, the visual distinction to solutions that were filtered out was insufficient, while DM2 would have liked a clearer connection between the axes in the plot and the objectives below. This feedback indicates that we should focus on improving the visual affordability and clarity of our tools, and make sure to provide sufficient documentation.

The DMs also suggested new features that would support their use of the system in practice. These included a comparison of the proposed schedules with the schedule currently active in the lab and a way to suggest objective weights that would result in a particular solution becoming the best.

Regarding the quantitative results, while the sample size of 2 is too small to produce statistically robust results, the achieved average SUS score of 80 (out of 100 maximum) is clearly above the average result of 68 (Sauro 2011). On an adjective-based scale for SUS results (Bangor, Kortum, and Miller 2009), it lies between "good" and "excellent". The scores are lowest for Q3 and Q4, which ask for ease of use. This also corresponds with the qualitative feedback from the DMs, as missing explanations and documentation was the main criticism. The individual responses for all questions of the SUS are included in the appendix (Section C).

Based on the feedback we received from the DMs, we recently added a comparison with the previously existing schedule which can be displayed in the PCP as well as textual suggestions for weight updates in the tooltips of the objective interaction indicators (d). This updated version of the

TLSP MO-Explorer is deployed in the lab of our industrial partner and used to support their daily scheduling activities.

## 6 Conclusions

In this paper, we have proposed a natural and intuitive method to explain the output of multi-objective optimization problems with different preference structures via Shapley regression values. These values provide insights into the interactions between objectives and suggestions on weight adjustments that are likely to lead to a desired outcome, while the low number of features compared to typical machine learning problems allows for their exact computation. We have applied our approach on the TLSP, a complex real-world scheduling problem, and empirically shown the effectiveness of the provided suggestions.

Additionally, we have introduced the TLSP MO-Explorer, a new decision support system for the TLSP, which consists of a graphical interface using parallel coordinate plots to visualize and explore populations of non-dominated solutions, combined with the explanations computed via Shapley regression values. We have demonstrated the usefulness of this system for domain experts in practice.

In the future, we would like to explore research directions suggested by the domain experts, in particular the automated setting of weights in order to achieve a desired solution profile. It would also be interesting to investigate how Shapley regression values perform for other problems, for example optimization problems using reference values like in the R-XIMO system.

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
