# OpenReview forum: "Preference Explanation and Decision Support for Multi-Objective Real-World Test Laboratory Scheduling"
_icaps-conference.org/ICAPS/2024/Conference — ICAPS 2024_

### Official Review · Reviewer_NhPk · 2024-01-18

**Significance And Importance:** 2
**Soundness:** 4
**Novelty:** 2
**Clarity:** 4
**Overall Evaluation:** 2
**Confidence:** 3

**Weaknesses:**

2: No major or minor weaknesses.

**Contributions Of The Paper:**

This short paper makes a number of contributions within the components of an integrated practical decision support system for multi-objective real-world test laboratory scheduling problems (TLSPs). Specifically, their contributions within the components of the system include:
1) Generalizing the eXplainable Interactive Multiobjective Optimization (XIMO) framework to multi-objective problems where the objective is to optimize a weighted sum.
2) Adapting this generalized XIMO problem to TLSPs.
3) Introducing a visualization system that allows for interactive updates by human users.
4) A user study to validate their proposed visualization system.

**Ethical Considerations:**

(1) Not Applicable: The paper does not have any ethical considerations to address

**Nomination For Best Paper:**

No

**Questions For Authors:**

Has the generalization of the XIMO framework to minimize a weighted sum been used or introduced in prior work? Or is that one of the novel contributions of this paper?

**Reproducibility:**

4: Authors promise to release code and domains (whichever apply).

**Strengths Of The Paper:**

The integration of the different components into a single practical decision support system and demonstrating its feasibility with real human users is noteworthy. The application to a complex real-world scheduling problem is also compelling.

**Weaknesses Of The Paper:**

Each of the contributions within each of the components are relatively incremental, and the user study includes only two partipants, which is not sufficient for statistical significance.

---

> ### Author Rebuttal · Authors · 2024-01-26
>
> - Number of user study participants: We are aware that the number of study participants is small (mentioned in lines 575ff in the paper).
>     The two DMs we interviewed are very experienced in the area of scheduling.
>     Unfortunately there are no other subject matter experts available at our industrial partner, so we were unable to obtain a larger sample size.
>     In light of this, the SUS scores should be taken as supplemental to the qualitative feedback.
>     We would also like to point out that the DMs in the study had not interacted with the MO-Explorer before the study, and they now use it in practice to support their daily scheduling activities.
>
> Questions:
> - XIMO with weighted sum objective: While the original introduction of XIMO by Misitano et al. allows for other preference structures, that paper exclusively deals with reference-points-based models.
>     As far as we know, there has been no other paper attempting to generalize this setting for other preference structures.
>     Thus, the XIMO setting for problems with weighted-sum-based preference structures is introduced for the first time in this paper.
>
>     Weight-based scalarization itself has long been a popular mechanism for multi-objective optimization, for an example see "Murata, T., Ishibuchi, H., \& Tanaka, H. (1996). Multi-objective genetic algorithm and its applications to flowshop scheduling. Computers \& industrial engineering, 30(4), 957-968".
>     We will add some more references for the final version of the paper.
>
>
> We would like to thank all reviewers for their positive comments and helpful remarks!

---

### Official Review · Reviewer_Ae1x · 2024-01-20

**Significance And Importance:** 2
**Soundness:** 4
**Novelty:** 3
**Clarity:** 3
**Overall Evaluation:** 2
**Confidence:** 4

**Weaknesses:**

1: Minor weaknesses that are easily fixable.

**Contributions Of The Paper:**

The paper presents an interactive tool to explain  the outcomes of multi-objective optimization problems featuring diverse preference structures, utilizing Shapley regression values. in order to make the paper self-contained, the authors provide a number of different contributions as background information, such as the Shapley regression values technique, and how it integrates within the Explainable Interactive Multi-objective Optimization (XIMO) paradigm, as well as providing a resolution framework for the Test Laboratory Scheduling Problem (TLSP), considered as the problem of reference. Successively, the paper applies the explanation system based on Shapley values to a number of Multi-objective TLSP instances. Finally, the results of these application provides the data upon which to present the visualization and interaction system (TLSP MO-Explorer) through which the users can visually analyze and iteratively modify the importance of the objectives preferences based on the system's responses, to the aim of fine tuning the obtained solutions towards the decision makers' desiderata. An analysis based on interviews with the user is also provided.

**Ethical Considerations:**

(1) Not Applicable: The paper does not have any ethical considerations to address

**Nomination For Best Paper:**

No

**Questions For Authors:**

1) Section 3, page 3, 2nd column, last paragraph (around line 270):
1a) In the pure PSA approach, how are the initial solutions found?
1b) is PVLNS to be intended similar to PSA-VI with the difference that the solutions are not handled to PSA as in PSA-VI but retained by VLNS for the whole process? In this case, the role of PSA in the process is not clear.

2) Section 3, page 4, lines (285-293): where do the results here described be read/seen?

3) Section 4, lines (378-382): Why is the case where the best solution is unchanged but the target value is reduced not considered a success?

POST-REBUTTAL COMMENTS
I thank the authors for their clarifying answers. I have no reason to change my original "accept" score, and in case the paper is accepted, I strongly encourage the authors to provide all the necessary modifications to address the reviewer's comments.

**Reproducibility:**

5: Code and domains (whichever apply) are already publicly available

**Strengths Of The Paper:**

The paper is well structured, and sufficienly clear to read (more about this in the Questions for authors section).
The paper presents an interesting solution for analysing the insights into the interplay among different objectives of TLSP multiobjective instances. The resulting visualization and interaction system seems a very informative and usable tool to offering valuable recommendations for weight adjustments that are likely to result in a desired outcome. Despite the overall work significantly exploits the results of different already existing contributions (in this sense, the authors' work mostly seem an effort of adaptation to the case at hand) the proposed system is sufficiently ingenious and meaningful. The paper is self-contained, and I have appreciated the authors' effort to provide all the necessary information for a complete understanding of all the system's building blocks, starting from the used technique to solve the multiobjective version of the TLSP (more about this in the next section of the review). Another strength of the paper is that the implemented visualization and interaction system seems at a rather mature usabilty level, providing a wide range of useful information.

**Weaknesses Of The Paper:**

The paper does not seem to have important weaknesses. From the structural standpoint, one small weakness is that the real objective of the study related to the various techniques to solve the TLSP (end of Section 3) remains a little obscure to me, because it seems a little "unrelated" to the rest of the paper, given also that the approach that is selected fot the successive analysis (PSA-VI) is not the best performing in terms of hypervolume. Maybe the solving approach of preference had to be the one that guarantees the highest level of solution diversity? If so, it should have been better specified.

---

> ### Author Rebuttal · Authors · 2024-01-26
>
> - Study of solution techniques: The main goal of this part of our work was to develop a multi-objective solution method that could produce a diverse set of (non-dominated) solutions for the analysis in the next section.
>     We selected PSA-VI for this, since it provided the best solution diversity among all evaluated approaches, while also achieving a high number of feasible solutions (in contrast to pure PSA, see lines 285ff and Figure 2).
>
>     For the purpose of exploring the solution space under different preference structures, high diversity is crucial as otherwise solutions along the extremes of the space would be ignored.
>     Once a satisfactory weight profile is found, a further optimization run could be performed to find the single best solution under that weight profile, using a different technique geared towards individual solution quality (e.g. (P)VLNS).
>
> Questions:
> - 1a) For PSA, the initial solutions are generated randomly (see line 257f). We also tried a greedy construction heuristic, but this did not improve the results.
> - 1b) Yes, this is the case. In PVLNS, (P)SA is not used at all. The main connection between these approaches is the weight update to move solutions away from their nearest neighbor, which is performed after each (VLNS) move, as in PSA, albeit with a higher update factor $\alpha$ to account for the lower number of moves performed by VLNS compared to SA.
> - 2 ) The detailed experimental results are included in Section B of the appendix, Tables 1-4. The results summarized in lines 285-293 are given in Table 2 of the appendix.
> - 3 ) If the best solution is unchanged, also its objective vector remains the same (including the value of the target objective), so this case cannot happen.

---

### Official Review · Reviewer_KgRH · 2024-01-23

**Significance And Importance:** 2
**Soundness:** 3
**Novelty:** 2
**Clarity:** 4
**Overall Evaluation:** 1
**Confidence:** 4

**Weaknesses:**

0: Minor weaknesses requiring some work to be addressed for the paper to be accepted.

**Contributions Of The Paper:**

This paper addresses the Test Laboratory Scheduling Problem, a multi-objective scheduling problem, with a hybrid Pareto-Simulated Annealing algorithm combined with Very Large Neighborhood Search. The paper presents a new method that uses Shapley values to suggest adjustments to the objective weights when generating new solutions. The paper also describes a new graphical tool TLSP MO-Explorer that allows decision makers to explore a population of solutions.

**Ethical Considerations:**

(2) Poor: The paper fails to address crucial ethical considerations

**Nomination For Best Paper:**

No

**Questions For Authors:**

How often did you apply the strategies used in the evaluation? Was it only once from w0 to w1? If so, wouldn't repeated applications further improve performance?

**Reproducibility:**

4: Authors promise to release code and domains (whichever apply).

**Strengths Of The Paper:**

The paper is well written and organized. It uses Shapley Values in a novel way that allows the solution to be visually explained and modified.

**Weaknesses Of The Paper:**

The evaluation of their method (PSA-VI) did not compare against other multi-objective optimization algorithms, instead it focused on different ablated versions of their own system. Similarly, it did not consider other strategies to use in the evaluation of their XIMO setup, such as those used by genetic and evolutionary algorithms for multi-objective optimization.

Secondly, the visualization and interaction portion of the paper likely fits better in an HCI context and would benefit from best practices from HCI and UI/UX design research.

Suggestions for ethical considerations:

Objectives for the TLSP:
1. None of the objectives address the impact of the solution on the environment and society. How could a decision maker consider the potential emissions, pollutants, waste, etc. from different solutions? How would different solutions impact the local community and economy?
2. Solutions to the TLSP also have a significant impact on the employees. A certain sequence of jobs could impact individual livelihoods, how much income they make, and their physical and mental health. See this report: https://www.ieai.sot.tum.de/wp-content/uploads/2022/06/Research-Brief_Algorithmic-Scheduling-in-Industry_Technical-and-Ethical-Aspects_June2022_FINAL.pdf

Transparency:
1. The tool that was designed is focused on conveying information to the decision makers, but the individuals who are the most impacted by the solutions have no visibility or understanding of how the schedules were produced and what factors were used to select the solution. This lack of transparency can lead to decreased employee morale and buy-in to the solution.
2. The constraints do not consider the employees' preferences and experience levels. Again, this reinforces a top-down application of the solution, with no consideration for individual employees and so it may result in a negative experience.

Visualization and Exploration Tool:

Although the tool builds on the PAVED system, it does not address important principles of inclusive HCI design and accessible technology. See the following:
1. Abascal, Julio, and Colette Nicolle. "Moving towards inclusive design guidelines for socially and ethically aware HCI." Interacting with computers 17.5 (2005): 484-505.
2. Fuchs, Christian, and Marianna Obrist. "HCI and society: towards a typology of universal design principles." Intl. Journal of Human–Computer Interaction 26.6 (2010): 638-656.
3. Miraz, Mahdi H., Maaruf Ali, and Peter S. Excell. "Adaptive user interfaces and universal usability through plasticity of user interface design." Computer Science Review 40 (2021): 100363.
4. Stumpf, Simone, et al. "Gender-inclusive HCI research and design: A conceptual review." Foundations and Trends® in Human–Computer Interaction 13.1 (2020): 1-69.

---

> ### Author Rebuttal · Authors · 2024-01-26
>
> - Comparison with other MOO approaches: We selected the MO solution approaches in this work based on the fact that SA and VLNS are the current state-of-the-art approaches for the TLSP (with single weighted-sum objective).
>     This enabled us to reuse and adapt well-performing algorithmic components described in earlier work to our proposed MO methods, as well as to perform a direct comparison of the results with those state-of-the-art single-objective methods.
>     In contrast, to the best of our knowledge there are no evolutionary or genetic algorithms for the TLSP on which we could have based our multi-objective approach.
>     In general, it seems very challenging to define effective crossover operators that do not introduce a large number of conflicts due to the many interconnected constraints of the TLSP.
>
>     To evaluate the effectiveness of the use of Shapley values for XIMO, we followed the same methodology as Misitano et al. (2022) introducing XIMO.
> - Ethical considerations: We did our best to identify and address all potential ethical concerns. If there are any concrete suggestions, we would be happy to improve the paper accordingly.
>
> Questions:
> - Number of strategy applications: For the purpose of this evaluation, we analyzed only the effect of a single application of each strategy (from $w^0$ to $w^1$) to show that following the suggestions computed from the Shapley regression values yields better results (in the sense of improving the target objective value) than the other options.
>     There is nothing conceptually different between the first and second iteration, except that the Shapley regression values would change after each application since they are relative to the current values of the preference structure.
>
>     It would certainly be possible to perform repeated weight updates for the same target objective, if reducing its value were the only concern.
>     Eventually this would lead to the target objective having a weight far above the others, effectively reducing the problem to a single-objective problem, which could be solved more efficiently by directly using a single-objective solver.
>     Instead, in practical application, we would expect DMs to adjust multiple weights iteratively (for different target objectives), until the values of all objectives match their subjective preferences.
>     As this behavior depends on a lot of factors and subjective appraisals it cannot reasonably be used as the basis for our empirical analysis.

---

### Meta-Review · Area_Chair_gUni · 2024-02-06

**Recommendation:** Accept (Oral)
**Confidence:** 5

**Metareview:**

Strengths:
- Complex real-world scheduling problem with multiple conflicting objectives.
- Using Shapley regression values in a novel way, to explain the interaction between objectives and suggest weight adjustments.
- Integration of different components into a practical decision support system and demonstrating feasibility with a short user study.
- Interesting visualization and interaction system -- one reviewer mentioned that it seems very usable and would provide valuable information and recommendations for weight adjustments, to result in solutions that match human user preferences.
- Paper is well written and organized.

Weaknesses:
- One of the reviewers included in their review comments and suggestions for ethical considerations. The final version of the paper must include an ethical statement tackling these comments and suggestions.
- Description should clarify that the preferred approach (PSA-VI) was chosen because it resulted in the best solution diversity among all the approaches evaluated, while also producing a high number of feasible solutions.
- The empirical evaluation did not compare the proposed approach against other multi-objective optimization algorithms.

**Ethical Considerations:**

(2) Poor: The paper fails to address crucial ethical considerations